# Molecular Identification of the Glutaredoxin 5 Gene That Plays Important Roles in Antioxidant Defense in *Arma chinensis* (Fallou)

**DOI:** 10.3390/insects15070537

**Published:** 2024-07-17

**Authors:** Qiaozhi Luo, Zhongjian Shen, Nipapan Kanjana, Xingkai Guo, Huihui Wu, Lisheng Zhang

**Affiliations:** 1School of Horticulture and Gardening, Tianjin Agricultural University, Tianjin 300392, China; 15704940113@163.com; 2State Key Laboratory for Biology of Plant Diseases and Insect Pests, Key Laboratory of Natural Enemy Insects, Ministry of Agriculture and Rural Affairs, Institute of Plant Protection, Chinese Academy of Agricultural Sciences, Beijing 100193, China; shenzhongjian@caas.cn (Z.S.); xingkaig0706@126.com (X.G.); 3Key Laboratory of Animal Biosafety Risk Prevention and Control (North), Ministry of Agriculture and Rural Affairs, Shanghai Veterinary Research Institute, Chinese Academy of Agricultural Sciences, Shanghai 200241, China

**Keywords:** *Arma chinensis*, *Grx5*, RNAi, VC content, CAT activity, hydrogen peroxide content

## Abstract

**Simple Summary:**

*Arma chinensis* (Fallou) is a predatory natural enemy that plays a vital role in the biological control of various agricultural and forestry pests. In this study, the temporal and spatial expression patterns of the *AcGrx5* gene in diapause and non-diapause conditions were quantified, and stress resistance was verified at low temperatures. A significant increase in metabolite content and antioxidant enzyme activity was found after the knockdown of the *AcGrx5* gene, which shows a new finding of genes in the diapause of *A. chinensis*.

**Abstract:**

Glutaredoxin (Grx) is a group of redox enzymes that control reactive oxygen species (ROS), traditionally defined as redox regulators. Recent research suggested that members of the *Grx* family may be involved in more biological processes than previously thought. Therefore, we cloned the *AcGrx5* gene and identified its role in *A. chinensis* diapause. Sequence analysis revealed the ORF of *AcGrx5* was 432 bp, encoding 143 amino acids, which was consistent with the homologous sequence of *Halyomorpha halys*. RT-qPCR results showed that *AcGrx5* expression was the highest in the head, and compared with non-diapause conditions, diapause conditions significantly increased the expression of *AcGrx5* in the developmental stages. Further, we found that 15 °C low-temperature stress significantly induced *AcGrx5* expression, and the expression of antioxidant enzyme genes *AcTrx2* and *AcTrx-like* were significantly increased after *AcGrx5* knockdown. Following *AcGrx5* silencing, there was a considerable rise in the levels of VC content, CAT activity, and hydrogen peroxide content, indicating that *A. chinensis* was exposed to high levels of reactive oxygen species. These results suggested that the *AcGrx5* gene may play a key role in antioxidant defense.

## 1. Introduction

Diapause is a state of growth and development induced by factors such as photoperiod or temperature in the environment, and many insects with hyper-diapause can survive in an unfavorable breeding season [1].

*Arma chinensis* (Fallou) belongs to Hemiptera, Atomidae, and Stuciidae. Few studies have been conducted on the diapause of the caterpillar. Only a few reports have reported the phenomenon of overwintering diapause in adults [2]. The environmental conditions, sensitive period of diapause induction, physiological and biochemical characteristics, and molecular mechanism of the diapause regulation of the parasite remain unclear. To investigate the physiological mechanism of the diapause of *A. chinensis*, we will contribute to further research on the basic theory of the diapause of *A. chinensis* and promote the industrial application of natural enemy insects. 

Reactive oxygen species (Ros) is produced as a byproduct of aerobic metabolism. As a signaling component, Ros is well known for its role in abiotic and biological stress-related events [3]. In many cases, Ros can cause severe oxidative stress and damage cellular proteins, lipids, or DNA [4,5]. Organisms can protect themselves from the toxicity of excess Ros by developing different effective mechanisms [6]. The Redox system in insects is mainly composed of the Thioredoxin (Trx) system and the Glutaredoxin (Grx) system. Glutathione ascorbate in the circulatory system plays an important role in the regulation of the redox state in insects. In addition, it can also regulate the redox state of the sulfhydryl group through two other systems so as to achieve the balance of the redox state in insects. The Thioredoxin system includes Thioredoxin (Trx), Triphosphopyridine nucleotide (NADPH), and Thioredoxin reductase (Tr), and the Glutathione system mainly includes NADPH, Glutathione reductase (Gr), Glutaredoxins (Grx), and Glutathione (GSH) [7,8].

In 1976, Holmgren found that glutaredoxin (Grx) could restore the growth and development of *Escherichia coli thioredoxin* (*Trx*) mutants [9]. *Grx* was originally thought to be an electron donor for Ribonucleotidereductase (RNR). Since then, *Grx* has been studied in *Escherichia coli*, yeast, mammals, and plants. *Grx* modifies the function of specific amino acid residues on proteins through reversible glutathione. It also maintains and regulates the cellular redox state and related signal pathways, participates in critical physiological activities such as electron transfer and iron metabolism in organisms, and plays a role in many important life activities such as biological oxidation stress and cell death [10]. There is little research on the insect *Grx* at home and abroad; only several insect *Grx* genes have been classified and identified, and their antioxidant function in oxidative stress has been preliminarily studied [11]. Moreover, in the model insect *Drosophila melanogaster*, it has been reported that the *Grx1* direct homologous gene CG6852 played a critical regulatory role in humans, which played a key role in heavy metal copper in the blood of *Drosophila melanogaster* [12]. In *Helicoverpa armigera*, a total of three *Grx* genes (*HaGrx*, *HaGrx3*, and *HaGrx5*) were identified. When *Helicoverpa armigera* is subjected to abnormal temperature stress and H_2_O_2_ treatment, *HaGrx* and the expression levels of *HaGrx3* and *HaGrx5* are changed; the authors speculated that they play an important role in protecting tissue from oxidative stress damage in the *Cotton Bollworm* [13]. However, in these studies, the basic biological characteristics and antioxidant mechanism of insect *Grx* have not been systematically and deeply studied.

In recent years, *Grx* has attracted a lot of interest due to its different and powerful features. In addition, *Grx* was reported to directly reduce H_2_O_2_ and dehydroascorbic acid (DHA), enhancing the ability of cells to clear reactive oxygen species [14]. Therefore, in this study, we measured hydrogen peroxide content after an *AcGrx5* gene knockout. We investigated the potential link between *Grx* and the diapause features of *A. chinensis* by using RNAi to knock down the expression of two components of this pathway. Examining the functions of the *AcGrx5* gene in regulating the diapause of *A. chinensis* can provide a theoretical basis for more in-depth knowledge of the molecular regulation of diapause.

## 2. Materials and Methods

### 2.1. Insect Rearing and Sample Preparation

The experimental larval *A. chinensis* was collected from the Langfang Base Laboratory of Plant Protection Institute, Chinese Academy of Agricultural Sciences, Hebei Province. The insect rearing was in a controlled room under the following conditions: temperature of 26 ± 5 °C; relative humidity of 60 ± 10%; photoperiod of a 16 h light/8 h dark cycle (normal developmental conditions). To induce diapause, newly emerged adults were transferred to an environment with the following conditions: 15/5 °C; 8 h light/16 h dark cycle; RH 60 ± 10% (diapause-inducing conditions). The *Bombyx mori* in the pupal stage was used to rear the stuck bug in succession to form a stable population for subsequent experiments. 

During the diapause phase and the sensitive period of diapause, the newly emerged adults were the most sensitive to diapause induction signals. The newly emerged adults were placed under the induced conditions, and they could enter diapause completely. All insects were collected, cleaned, and frozen using liquid nitrogen. The samples were stored at −80 °C until the next analysis. Each treatment was conducted with three replicates, each comprising only females.

### 2.2. Gene Cloning and Sequence Analysis

One fragment encoding the putative Glutaredoxin domain-containing cysteine-rich protein and Glutaredoxin-related protein was identified based on the transcriptome database that was previously established in our laboratory. Total RNA was extracted from individual *A. chinensis* using the TRIzol Reagent (Invitrogen, Carlsbad, CA, USA), following the manufacturer’s instructions. The specific primers (Table 1) that were used for the polymerase chain reaction (PCR) amplification were designed using DNAMAN v.6.03 software (Lynnon Biosoft, San Ramon, CA, USA) and synthesized by the TSINGKE Biological Technology Company (Beijing, China). PCR was conducted with a 2 × Taq Plus Master Mix (Vazyme). The target genes were subcloned into the pEASY-T5 Cloning Vector (Transgen, Beijing, China) and sequenced by the TSINGKE Biological Technology Company. The PCR products of the expected size were excised from the gels and transformed using the In-Fusion HD Cloning Kit (Takara Biotechnology Co., Ltd., Dalian, Liaoning, China) before being sequenced. Finally, the 3′UTR, 5′UTR, and coding DNA sequence (CDS) of the *AcGrx5* gene were obtained.

The full-length cDNA of *AcGrx5* was identified using the National Center for Biotechnology Information (NCBI) web server. The amino acid sequences of *AcGrx5* were determined using the ExPASy Translate tool (https://web.expasy.org/translate/ accessed on 23 March 2024), and the physiochemical features of the proteins were predicted using ExPASy Protparam (https://web.expasy.org/protparam/ accessed on 23 March 2024). To investigate the evolutionary relationships of *AcGrx5*, the *AcGrx5* amino sequences of various species were downloaded from the NCBI and aligned with sequences of *AcGrx5* using the ClustalW 2 and ESPript 3.0 webservers. To analyze protein sequence similarities and secondary structure information, a model of the *AcGrx5* protein was constructed using SWISS-MODEL (https://swissmodel.expasy.org/, accessed on 23 March 2024). The phylogenetic tree was constructed using the neighbor-joining (NJ) method and plotted with Molecular Evolutionary Genetics Analysis (MEGA) version 6.0 software. Bootstrap sampling was performed with 1000 replicates [15].

### 2.3. mRNA Expression by RT-PCR Analysis

The mRNA expression in all samples collected in this experiment was detected by quantitative real-time PCR (qRT-PCR). Total RNA was extracted from the insect samples using the TRIzol Reagent (Invitrogen, Carlsbad, CA, USA), and 1 μg of total RNA was reverse transcribed to cDNA using TransScript^®^One-Step gDNA Removal and cDNA Synthesis SuperMix (TransGen Biotech, Beijing, China) under the following conditions: 42 °C for 30 min and 85 °C for 5 s. Next, real-time PCR was performed using a TOROGreen^®^ 5G qPCR Premix (Toroid Technology Limited, Changzhou, China) and a LightCycler^®^ 96 Instrument (Roche, Rotkreuz, Switzerland). The program was run in three steps: 95 °C for 5 min, followed by 40 cycles at 98 °C for 10 s and 53 °C for 20 s. A dissociation step cycle (95 °C for 10 s, 65 °C for 60 s, and 97 °C for 1 s) was added for the melting curve analysis. All reactions were run in triplicates, the PCR amplification was performed in 20 μL, and each reaction consisted of 10 μL of TOROGreen Premix, 0.8 μL of each specific primer (Table 1), 1 μL of sample cDNA, and 7.4 μL of nuclease-free water. When the reactions were complete, CT values were determined using fixed threshold settings. A total of three independent biological samples were included for each group, and three technical replicates of each biological sample were processed for all reactions. The PCR data were analyzed using the 2^−ΔΔCT^ method. Statistical analyses were conducted using IBM SPSS statistics 26.0. A one-way ANOVA, followed by Tukey’s multiple range test, estimated the statistical differences. When *p* < 0.05, the difference was considered significant. 

### 2.4. Tissue-Specific Expression of AcGrx5 mRNA

In order to assess the expression of *AcGrx5* in different tissues of *A. chinensis*, quantitative real-time PCR (qRT-PCR) was conducted using the RNA of the head, ovary, fat body, midgut, and malpighian tubule from thirty healthy *A. chinensis*.

Newly emerged (within 24 h) female adults were placed under normal development conditions and diapause induction conditions, respectively. After 10 days, 15 adults were collected for morphological and anatomical observation and photography. The anatomical process was as follows: Firstly, the feet, wings, mouthparts, and antennae were cut off an adult caterpillar with scissors. Then, the abdominal edge line was cut completely with scissors, and the abdominal epidermis was cut along the border of the chest and abdomen. Secondly, the treated stinkbug was placed on a wax tray and secured with an insect needle inserted at a 45 °C angle at the edge of the chest, and an appropriate amount of phosphate buffer was poured into the wax dish and observed under a type of microscope. Thirdly, tweezers were used to separate the bottom membrane of the abdomen from the epidermis, and the epidermis was removed. After the abdominal fat body was cleaned, the inner genitalia were completely removed and placed into a clean PBS solution. Finally, an appropriate amount of PBS solution was dropped onto the ultra-depth field platform, each tissue was placed in it, and photos were taken in turn.

### 2.5. Low Temperature Treatment

The newly emerged adults were placed under normal feeding conditions of a 16 h light/8 h dark cycle, and three temperature gradients were set: 5 °C, 15 °C, and 15/5 °C. After 24 h, 48 h, and 72 h, the samples were taken, and individual treatments were performed with four replicates; for each replicate, only three females were chosen in this study.

### 2.6. Gene Expression Was Compared between Diapause Induction and Normal Developmental Stages

Samples from non-diapause 0 days, 3 days, 6 days, 9 days, and 12 days, and diapause 10 days, 20 days, 30 days, and 40 days were collected. The insects that were injected with dsAcGrx5 and dsAcGFP were reared in pairs under normal developmental conditions. Samples were taken 48 h later. All insects were cleaned and then frozen using liquid nitrogen. The samples were then stored in a refrigerator (−80 °C) until analysis was performed. Each treatment was performed with three biological replicates, and each replicate consisted of three females.

### 2.7. RNA Interference (RNAi) 

To investigate the function of *AcGrx5* during the diapause of *A. chinensis*, RNAi was used to suppress gene expression in female adults of *A. chinensis*. The 301 bp regions of *AcGrx5* were amplified from cDNA using the PCR technique with the specific primers for dsAcGrx5. The dsRNAs were obtained using the MEGAscript T7 High Yield Transcription Kit (Invitrogen, Carlsbad, CA, USA). Double-stranded RNA (dsRNA) against green fluorescent protein (GFP) was used as the control. The dsRNA injection was administered to females of similar size, after which they were reared for one day (after eclosion) under diapause-inducing conditions. Each female had 1 μL (2 μg/μL) of dsRNA solution for each target gene injection administered at the internode membranes between the second and third abdominal segments. The control groups were injected with an equivalent amount of dsGFP solution. After injection, whole insects were collected for subsequent RT-qPCR experiments.

To see if knocking out *AcGrx5* causes changes in other related genes, in all RNAi experiments, females reared for one day (after adult eclosion) were injected with dsRNA. The insects that were injected with dsAcGrx5 and dsAcGFP were reared in pairs under normal developmental conditions. In RT-qPCR experiments, the transcript abundances of related genes were examined two days after RNAi-injection (dsAcGrx5 and dsAcGFP). Each treatment was performed with three biological replicates, with each replicate containing four females. 

### 2.8. Quantification of Antioxidant Genes after AcGrx5 Silencing

QRT-PCR was performed to analyze the relative expression levels of the antioxidant genes, such as *AcTrx2, AcTrx-like, AcPDI*, and *AcGrxcr*, in the shrimp of *AcGrx5*-silenced and GFP groups. All gene-specific primer pairs are listed in Table 1.

### 2.9. Comparison of Each Oxide Content after Interference

The insects that were injected with dsAcGrx5 and dsAcGFP were reared in pairs under normal developmental conditions. Two days later, the sample was collected, and the experiment was conducted. The BCA protein concentration detection kit used was from the Biyuntian Biotechnology Company. Firstly, the protein standards and BCA working fluids were prepared. Then, the protein concentration was measured. The standard product was added to the standard product holes of the 96-well plate by 0, 1, 2, 4, 8, 12, 16, and 20 μL. A total of 200 μL of BCA working liquid was added to each hole and incubated at 37 °C for 30 min. The absorbance of A562 was measured using an enzymoleter, and the protein concentration of the sample was calculated according to the standard curve and the sample volume used. For the VC content, CAT activity, analysis of the superoxide dismutase activity, and detection of H_2_O_2_, all the test kits were bought from Nanjing Company. For the tissue to be measured accurately, 9 times the volume of normal saline was added, centrifuged at 2500 RPM for 10 min, and the supernatant to be measured was collected. According to the instructions, the reagent was added in turn, and the absorbance value of each tube was determined according to the respective wavelength. 

### 2.10. Statistical Analysis 

The GraphPad Prism 9.0 (GraphPad Software, San Diego, CA, USA) was used for all statistical analysis. The quantitative real-time PCR analysis of gene relative expression at different stages was also statistically evaluated using a one-way ANOVA analysis. The Students’ test and Gehan–Breslow–Wilcoxon test were used to compare, for the different tissues from the diapause and non-diapause, the gene expression and jamming efficiency between the treatment and control groups in RNAi experiments (*p* < 0.05). The SPSS 27 software was used to perform statistical analysis of the experimental data, and the values were represented as the means ± the standard error of the mean (SEM) with at least three independent repetitions.

## 3. Results

### 3.1. The Identification of AcGrx5 Genes and Phylogenetic Analysis of AcGrx5 Proteins

Sequence analysis showed that the total length of the open reading frame (ORF) was 432 bp, encoding 143 amino acids. The results of the protein’s physical and chemical properties based on the amino acid sequence of *AcGrx5* indicated that the predicted protein molecular formula was C_1333_H_2236_N_432_O_567_S_50_, the molecular mass was 35 KD, and the theoretical isoelectric point was 5.33 (Figure 1). In comparison with *Grx* homologous sequences from different insect species, *AcGrx5* was found to contain conserved forked head domains (Figure 1). Phylogenetic tree results revealed that *AcGrx5* belonged to an evolutionary clade with the proteins of other species and was closely related to the Grx5 protein from *Halyomorpha halys* (Figure 2).

### 3.2. Expression Profiles of AcGrx5 at Different Stages

RT-qPCR was used to investigate the transcriptional profiles of *AcGrx5* at different developmental stages in *A. chinensis* under non-diapause and diapause conditions. The temporal expression profile showed that the *AcGrx5* mRNA was abundant in the NE stage and was low in other stages (Figure 3A). However, under diapause conditions, *AcGrx5* expression was increased significantly at various developmental stages, and *AcGrx5* expression was increased 3.26 times in the D40 stage compared with the NE stage (Figure 3B). Thus, these results suggest that *AcGrx5* may play an important role in the *A. chinensis* diapause.

### 3.3. Expression Profiles of AcGrx5 at Different Tissues

As shown in Figure 4, significant differences were observed in the expression patterns of *AcGrx5* in various tissues, and in both non-diapause and diapause conditions, the gene expression was highest in the head. In non-diapause conditions, *AcGrx5* expression was lowest in the fat body and higher in the midgut than in the ovary and malpighian tubule (Figure 4A). However, the expression level of *AcGrx5* was lowest in the midgut and higher in the fat body and the ovary under diapause conditions (Figure 4B). These results suggest that diapause conditions regulated *AcGrx5* expression in different tissues.

### 3.4. Expression Profiles of AcGrx5 at Different Temperatures and Different Times

To verify that the *AcGrx5* gene has a significant response to low-temperature stress, we investigated the expression levels of the *AcGrx5* gene at 26 °C, 5 °C, 15 °C, and 5/15 °C for 24, 48, and 72 h. The RT-PCR results showed that *AcGrx5* expression was significantly induced at 15 °C at all time points, compared with 26 °C, while 5 °C and 5/15 °C had little effect on the expression of *AcGrx5* (Figure 5). The results showed that the expression level of the *AcGrx5* gene was significantly different under different temperature pressures.

### 3.5. Expression Profiles of Other Antioxidant Genes after AcGrx5 Silencing

RT-qPCR was used to detect the expression of other antioxidant genes after *AcGrx5* was knocked down. Compared with the control, the mRNA levels of *AcGrx5* were reduced by 57% after dsRNA injection. After *AcGrx5 silencing*, the relative level of *AcTrx2* and *AcTrx-like* was upregulated significantly, and the relative level of *AcPDI* and *AcGrxcr* showed no significant difference. The results revealed complex interactions among some family members (Figure 6).

### 3.6. Assay of Antioxidant Enzyme Activities and Metabolite Contents after AcGrx5 Silencing

RNAi was used to silence *AcGrx5* expression in non-diapausing females. Compared with the control, the VC content was increased by two times (Figure 7A). As shown in Figure 7B, CAT activity increased significantly at 48 h after dsGrx5 injection, while SOD activity decreased significantly (Figure 7C). Similarly, the H_2_O_2_ content increased after *AcGrx5* knockdown (Figure 7D), suggesting that *AcGrx5* was involved in the removal of reactive oxygen species.

## 4. Discussion

Diapause is defined as a period of developmental stagnation during which an organism needs to overcome many stressful factors [16,17]. The objective of this study was to further understand the molecular basis of stress resistance of *A. chinensis* during diapause by analyzing the relative gene expression of *A. chinensis*. *Grx5*, a small molecule of the monomercaptan *Grx*, has previously been studied in bacteria, yeast, *Arabidopsis*, *zebrafish*, and humans [18,19,20]. As a redox enzyme, *Grx* is involved in the maintenance and regulation of intercellular redox balance. Current research on *Grx* genes has focused on plants and bacteria, and most of the genes have been cloned to complete expression analysis. Zhang et al. found that *Grx*, *Grx3*, and *Grx5* genes of *Helicoverpa armiger* were induced by different temperatures and H_2_O_2_; they subsequently detected that these genes had antioxidant defense functions [13]. An et al. found that the *Grx2* gene was rapidly induced to express when *Ostrinia furnacalis* was subjected to starvation, ultraviolet light exposure, and high-temperature stress [21]. Shen et al. found that after *Trx2* and *Trx-like1* gene knockouts, POD enzyme activity, hydrogen peroxide, and ascorbic acid metabolites content increased [22]. Yao et al. found that the *Grx1* and *Grx2* genes of *Apis cerana cerana* were induced by extreme factors such as high and low temperatures, H_2_O_2_, insecticides, and mercury [23].

In the non-diapause period, NE showed the highest gene expression, indicating that the newly emerged adult was most sensitive to signal stimulation. It has been hypothesized that *AcGrx5*, as a monomercaptan of *Grx*, may be involved in the assembly and transfer of iron–sulfur clusters during the early development of *A. chinensis*, thus contributing to its smooth development [22]. With the increase of diapause days, the expression of the *AcGrx5* gene increased gradually in the diapause period. Similarly, the upregulation of *Trx* has been observed in the bumblebee *Bombus ignites* in response to low temperatures [24]. In addition, it was also recently discovered that the upregulation of *Trx* in the first few months of diapause in *O. nubilalis* may be due to the increased production of hydrogen peroxide.

*AcGrx5* is expressed in all tissues, indicating that it has a wide range of biological functions, but the expression varies greatly in different tissues [25]. However, the expression of *AcGrx5* in different tissues was very different. The expression of *AcGrx5* was the highest in the head during non-diapause and diapause. This suggested that *AcGrx5* may play an important role in sensing photoperiodic signals [26].

Temperatures at different levels can change the photoperiodic induction of diapause. Although in most models on the effect of temperature on diapause induction, the temperature only acted along with the photoperiodic response; the temperature can drive diapause induction independently of the photoperiodic response [27]. In *Andrallus spinidens*, a temperature drop to 22.5 °C can cause it to enter diapause without a photoperiod [28]. In this experiment, with 26 °C as the control, three different temperatures were set at 5 °C, 15 °C, and 5/15 °C to explore the influence of temperature on the *AcGrx5* gene. The results showed that *AcGrx5* gene expression was higher at 15 °C than at other temperatures. The gradual increase in the relative abundance of mRNA analyzed during cold-acclimated larval diapause may reflect the effect of low temperature on gene expression and mRNA stability, which has been documented in previous dormancy [17].

RNAi is a biological process in which double-stranded RNA (dsRNA) induces sequence-specific gene silencing by the targeted degradation of RNA. As a tool to knock down the expression of individual genes after transcription, RNAi has been widely used to study the cellular function of genes [29,30]. This experiment wanted to explore whether interfering with the *AcGrx5* gene affected the expression of other related genes. The results showed that only the *Trx-like* gene was significantly upregulated after interference with the *AcGrx5* gene. Therefore, *Trx-like* may play an important role in the antioxidant reaction to cold stress [31]. In addition, in order to better verify the defense mechanism of *AcGrx5* in response to low-temperature stress in a diapause environment, we used dsRNA to significantly inhibit the expression of *AcGrx5*.

The study found that chronic intermittent hypoxia (CIH), through the Wnt/β-catenin pathway and the adjustable *Grx1* list, exacerbates skeletal muscle aging [32]. Therefore, it was inferred that CIH causes oxidative stress in skeletal muscle cells and finally reduces the expression of *Grx1*, which then intensifies oxidative stress and leads to the apoptosis of skeletal muscle cells. Studies have also shown that a *Grx1* knockout can promote the stability of HIF, inhibit the Wnt/β-catenin pathway, reduce oxidative stress, promote pulmonary angiogenesis and alveolar formation, and improve acute lung injury caused by hyperoxia [33]. In summary, *Grx1* plays an important role in the occurrence and development of various diseases caused by oxidative stress. Therefore, it is of great significance to further study the pathogenesis of the Grx family and oxidative stress in the above diseases.

## 5. Conclusions

In summary, we identified and determined the temporal and spatial expression patterns of the *AcGrx5* gene in *A. chinensis* under diapause and non-diapause conditions. We found that the gene expression was the highest at 40 days of diapause, indicating that the gene was indeed resistant to low-temperature stress. After the *AcGrx5* gene was knocked out, the VC content, CAT activity, and hydrogen peroxide content were increased, which further indicated that the *AcGrx5* gene has an oxidative stress function. Taken together, our findings suggested that the *AcGrx5* gene may play a key role in antioxidant defense.

## Figures and Tables

**Figure 1 insects-15-00537-f001:**
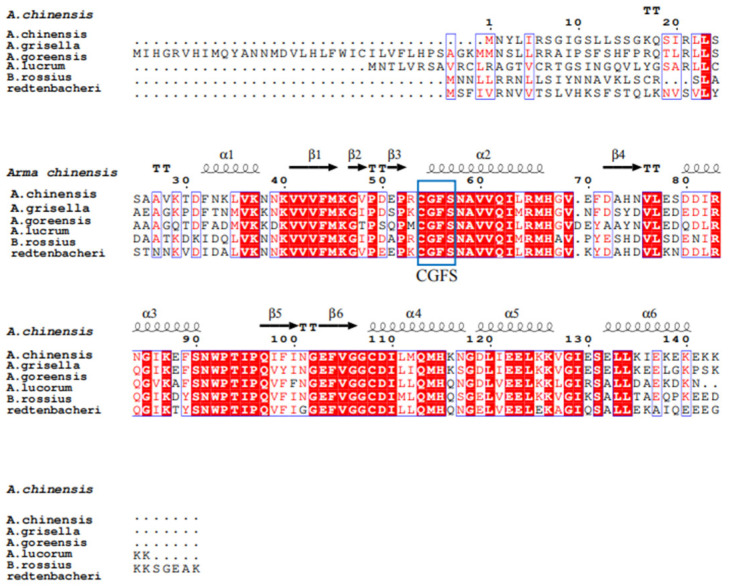
*A. chinensis* and corresponding proteins from other insect species. Red regions indicate conserved amino acid residues. Red letters indicate similar residues. Conserved active sites C-G-F-S are highlighted in a blue box. The results of the amino acid sequence counts are shown at the top of each row. The GenBank login numbers are as follows: *A. chinensis* (present study), *Achroia grisella* (XP_059062949.1), *Amyelois transitella* (XP_013194954.2), *Apolygus lucorum* (KAF6204643.1), and *Bacillus rossius redtenbacheri* (XP_063217809.1). Labeling: α, β, η, and TT marked alpha-helices, eta-helices, beta strands, and beta turns, respectively.

**Figure 2 insects-15-00537-f002:**
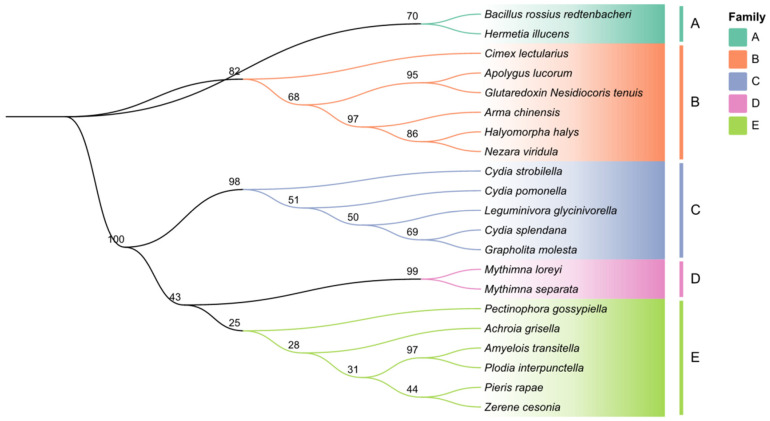
Phylogenetic tree of the Grx5 protein from *A. chinensis* and other insects. The phylogenetic tree was constructed based on the amino acid sequences of Grx5 using the WAG-based neighbor-joining method with 1000 bootstraps. The families A, B, C, D, and E represent the ptera order of insects. Family B: *Hemiptera*, and Family A, C, D, E: *Lepidoptera*.

**Figure 3 insects-15-00537-f003:**
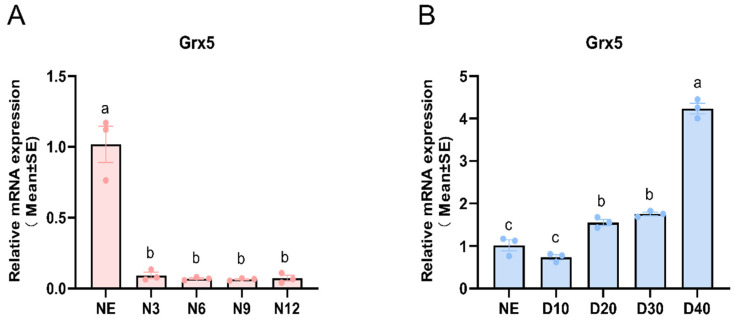
Expression patterns of *AcGrx5* at different developmental stages under non-diapause (**A**) and diapause conditions (**B**). NE: newly emerged females, N3, N6, N9, and N12; females were reared for 3, 6, 9, and 12 days under non-diapause conditions, respectively. D10, D20, D30, D40: females were reared for 10, 20, 30, and 40 days under diapause conditions, respectively. Data are presented as the mean ± SE of triplicate biological replicates. Statistical differences between different tissues are represented with different letters using a one-way analysis of variance (ANOVA) with Tukey’s multiple comparisons test; *p* < 0.05.

**Figure 4 insects-15-00537-f004:**
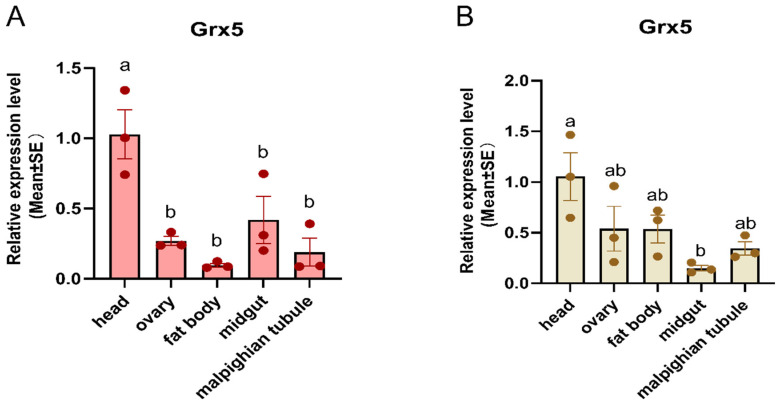
Expression profiles of *AcGrx5* in different tissues under non-diapause (**A**) and diapause conditions (**B**). The tissue samples included the head, fat body, midgut, malpighian tubule, and ovary of a female adult reared for 10 days under diapause and non-diapause conditions. Data are presented as the mean ± SE of triplicate biological replicates. Statistical differences between different tissues are represented with different letters using a one-way analysis of variance (ANOVA) with Tukey’s multiple comparisons test; *p* < 0.05.

**Figure 5 insects-15-00537-f005:**
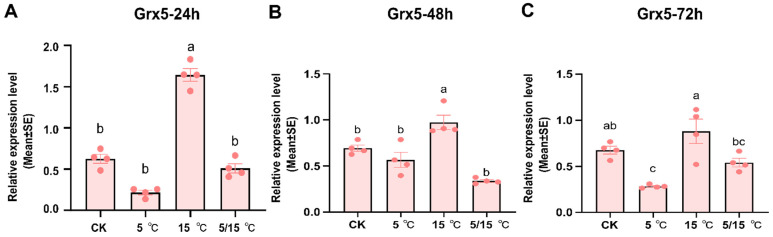
Expression profiles of *AcGrx5* at different temperatures at 24 h (**A**), 48 h (**B**), and 72 h (**C**). Data are presented as the mean ± SE of triplicate biological replicates. Statistical differences between different temperatures are represented with different letters using a one-way analysis of variance (ANOVA).

**Figure 6 insects-15-00537-f006:**
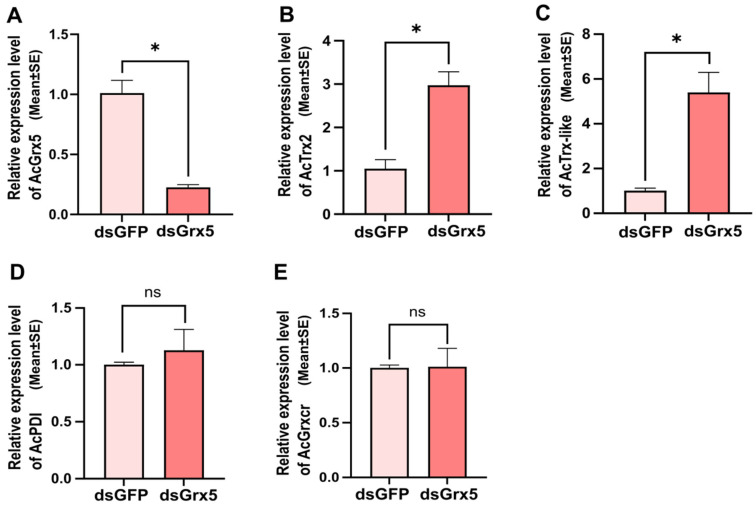
Expression analysis of other antioxidant genes after *AcGrx5* silencing. Relative expression of *AcGrx5* (**A**), *Trx2* (**B**), *Trx-like* (**C**), *PDI* (**D**), and *Grxcr* (**E**) (*n* = 16 for dsGFP, *n* = 16 for dsGrx5) after injection with dsRNA. Data are presented as the mean ± SE. Asterisk indicates significant differences (*t*-test; * *p* < 0.05); ns: no significance. *Trx2*: thioredoxin 2, *Trx-like*: Tioredoxin-like, *PDI*: Protein disulfide isomerases, *Grxcr*: glutaredoxin domain-containing cysteine-rich protein.

**Figure 7 insects-15-00537-f007:**
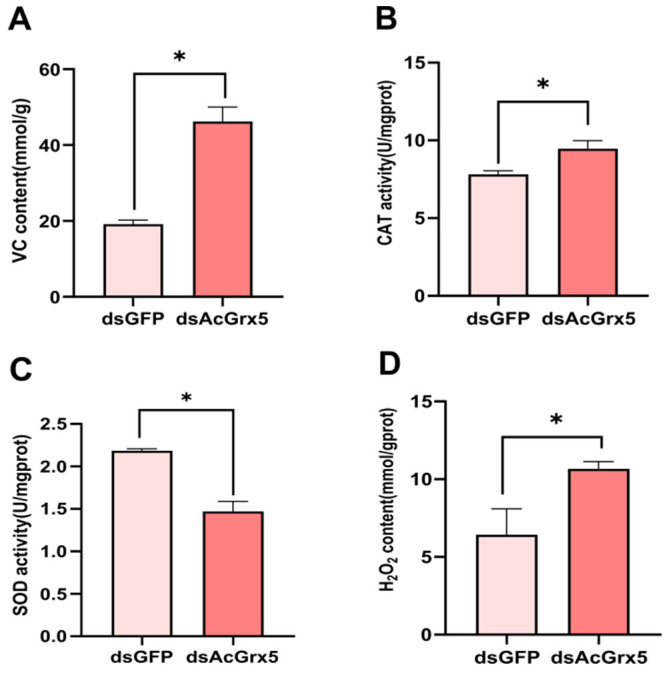
Effects of *AcGrx5* knockdown on metabolite contents and enzymatic activity. The VC content (**A**), the activity of CAT (**B**), the SOD activity (**C**), and the H_2_O_2_ content (**D**) were measured after injection with dsRNA. Data are presented as the mean ± SE. Asterisk indicates significant differences (*t*-test; * *p* < 0.05).

**Table 1 insects-15-00537-t001:** Summary of primers used in this study.

Primers	Sequences	Tm/°C	Product Size/bp
AcGrx5-F	TAGCTGCCCAACGTTGATAAC	58	779
AcGrx5-R	TGGTGTACTCTTCAGTAAACA	58	779
qAcGrx5-F	ATGAATTATTTGATTAGGTC	53	82
qAcGrx5-R	CCGCAGCACTTGAAAGTAGTCT	53	82
dsAcGrx5-F	T7-TATGAAAGGTGTCCCAGATGA	60	301
dsAcGrx5-R	T7-CTATTTCTTCTCCTTTTCT	60	301
qAcTrx2-F	ATGGTTTCTTTTCATTTTC	53	120
qAcTrx2-R	AATTTTGCGGCAAGGACC	53	120
qAcTrx-like-F	ATGGCACTTTCAGTTTTAATCG	53	84
qAcTrx-like-R	GCATATGAGCCGGTTGCAT	53	84
qAcPDI-F	TCAAGGTGACAGGACAAAGG	53	86
qAcPDI-R	TTTCTTGGCGTGTTATTTGC	53	86
RPL27-F	CCACCTTAGACGTAACCAGAA	53	116
RPL27-R	ACAGTCGATAACTGGTGCCT	53	116

## Data Availability

All data are provided within the text.

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
