# Peer review of "Molecular Identification of the Glutaredoxin 5 Gene That Plays Important Roles in Antioxidant Defense in Arma chinensis (Fallou)"

_insects, 2024, doi:10.3390/insects15070537_

Round 1

Reviewer 1 Report

Comments and Suggestions for Authors

This research paper shows how a Redox Regulator gene, Grx5 plays an important role in antioxidant defense. The study is done comparing both when insects are in diapause and non-diapause condition.

Some minor changes:

Table 1: Add Tm and amplicon size for each primer

Author Response

Dear reviewer:

I am very grateful to your comments for the manuscript. According with your advice, we added Tm and amplicon size for each primer.

Primers Sequences

Tm/°C

Product size/bp

AcGrx5-F TAGCTGCCCAACGTTGATAAC

58

779

AcGrx5-R TGGTGTACTCTTCAGTAAACA

58

779

qAcGrx5-F ATGAATTATTTGATTAGGTC

53

82

qAcGrx5-R CCGCAGCACTTGAAAGTAGTCT

53

82

dsAcGrx5-F T7-TATGAAAGGTGTCCCAGATGA

60

301

dsAcGrx5-R T7-CTATTTCTTCTCCTTTTCT

60

301

qAcTrx2-F ATGGTTTCTTTTCATTTTC

53

120

qAcTrx2-R AATTTTGCGGCAAGGACC

53

120

qAcTrx-like-F ATGGCACTTTCAGTTTTAATCG

53

84

qAcTrx-like-R GCATATGAGCCGGTTGCAT

53

84

qAcPDI-F TCAAGGTGACAGGACAAAGG

53

86

qAcPDI-R TTTCTTGGCGTGTTATTTGC

53

86

27-F CCACCTTAGACGTAACCAGAA

53

116

27-R ACAGTCGATAACTGGTGCCT

53

116

Reviewer 2 Report

Comments and Suggestions for Authors

In the manuscript titled “Molecular Identification of Glutaredoxin 5 gene that Plays Important Roles in Antioxidant Defense in Arma chinensis, the temporal and spatial expression pattern of the AcGrx5 gene was quantified under diapause and non-diapause conditions. Additionally, the AcGrx5 gene was silenced to measure related metabolites. This research explores the function of the AcGrx5 gene during diapause, offering a theoretical foundation for a more profound comprehension of the molecular regulation of diapause. However, there are concerns regarding the experimental conclusions and article writing that need to be addressed. I kindly request you to provide explanations and make necessary modifications.

I think this manuscript is suitable for publication in the Insects, and revision should be recommended.

COMMENTS:

1.After AcGrx5 gene was knocked down, VC content, CAT activity and hydrogen peroxide content were increased. Compared with the control group, the SOD content was reduced, but you did not discuss this result. What is the reason? Is there any literature that can prove this result?

2.In line 17, is the knockdown better than knockout ?

3.In this article, I think you have a problem with the Latin writing of Stuck bug. Firstly, in line 13, you did not add a Namer after Latin. In line 18, Arma chinensis should be abbreviation. In addition, in line 38, the Namer should not be italicized. Finally, I advise you add the family and genus of A. chinensis in your title.

4.In line 39, Why is “caterpillar” italicized?

5.In line 94, bread is usually used in plants, please use rear instead.

6.In Figure 3, the font size of the horizontal and vertical coordinates is not uniform. In addition, I think it is best to notably label diapause and non-diapause in the combined diagram. The same as Figure 4.

7.In Figure 5, I didn’t find your method for post-hoc analysis.

Author Response

Dear reviewer:

I am very grateful to your comments for the manuscript. According with your advice, we amended the relevant part in the manuscript. Some of your questions were answered below.

1.After AcGrx5 gene was knocked down, VC content, CAT activity and hydrogen peroxide content were increased. Compared with the control group, the SOD content was reduced, but you did not discuss this result. What is the reason? Is there any literature that can prove this result?

Response: We think that the decrease in SOD content may be due to the increase in the content of other antioxidant enzymes, which is a compensation for the maintenance of balance in the organism. Because in Grapholita molesta, after knocking down GmTpx, the enzymatic activities of superoxide dismutase (SOD) was also reduced.

[1] Liu Y.; Zhu F.; Shen Z.; Moural TW.; Liu L.; Li Z.; Liu X.; Xu H. Glutaredoxins and thioredoxin peroxidase involved in defense of emamectin benzoate induced oxidative stress in Grapholita molesta. Pestic Biochem Physiol. 2021, 176, 104881. http://.doi: 10.1016/j.pestbp.2021.104881

2.In line 17, is the knockdown better than knockout?

Response: We agree with the comment and rewrite it in the revised manuscript.

3.In this article, I think you have a problem with the Latin writing of Stuck bug. Firstly, in line 13, you did not add a Namer after Latin. In line 18, Arma chinensis should be abbreviation. In addition, in line 38, the Namer should not be italicized. Finally, I advise you add the family and genus of A. chinensis in your title.

Response: We agree with the comment and rewrite them in the revised manuscript.

4.In line 39, Why is “caterpillar” italicized?

Response: We apologize for the language problem in the original manuscript. We have been modified it.

5.In line 94, bread is usually used in plants, please use rear instead.

Response: We agree with the comment and rewrite it in the revised manuscript.

6.In Figure 3, the font size of the horizontal and vertical coordinates is not uniform. In addition, I think it is best to notably label diapause and non-diapause in the combined diagram. The same as Figure 4.

Response: We are grateful for the suggestion. As suggested by the reviewer, we have modified Figure 3 to distinguish between diapause and non-diapause.

7.In Figure 5, I didn’t find your method for post-hoc analysis.

Response: We feel sorry for our carelessness. The method of post-hoc analysis was not used, we have already deleted it in the revised manuscript.

Reviewer 3 Report

Comments and Suggestions for Authors

Manuscript Title: -

Molecular Identification of Glutaredoxin 5 gene that Plays Important Roles in Antioxidant Defense in Arma chinensis

The research provides a thorough and clear description of the experimental procedures, including mRNA expression analysis and gene cloning, which helps in understanding the research approach and replicating the study. The introduction and literature review sections offer a solid foundation, effectively summarizing the role of reactive oxygen species and the redox system in insect physiology, which sets the stage for the study. The research has practical implications for biological control in agriculture, highlighting the importance of understanding genetic and physiological mechanisms in natural enemy insects like Arma chinensis. Some of the minor suggestions before acceptance: -

1.      Conduct a thorough review for minor grammatical and spelling errors. For example, ensure subject-verb agreement and correct use of tenses.

2.      Figures 1 and 2 are informative, but adding more detailed legends and annotations, particularly highlighting the evolutionary significance in the phylogenetic tree, would make the figures more impactful and easier to interpret.

3.      The first time an abbreviation is used, such as "ROS" for reactive oxygen species, it should be spelled out in full with the abbreviation in parentheses. Ensure all abbreviations are defined when first introduced.

4.      The discussion is comprehensive, but integrating more recent studies would provide a better context for the research.

5.      Check the formatting of the references section to ensure consistency with the journal’s guidelines. Pay attention to punctuation, italics for species names, and correct citation order.

6.      Make sure all units of measurement and symbols (e.g., μg/mL, °C, p-values) are consistently formatted and follow the same style throughout the manuscript.

7.      Mention the software and version used for statistical analysis or any other computational work. This adds to the reproducibility of the study.

Comments on the Quality of English Language

Minor typo- and grammar corrections required. 

Author Response

Dear reviewer:

I am very grateful to your comments for the manuscript. According with your advice, we amended the relevant part in the manuscript. Some of your questions were answered below.

1.Conduct a thorough review for minor grammatical and spelling errors. For example, ensure subject-verb agreement and correct use of tenses.

Response: We feel sorry for our carelessness. In our resubmitted manuscript, the type is revised. Thanks for your correction.

2.Figures 1 and 2 are informative, but adding more detailed legends and annotations, particularly highlighting the evolutionary significance in the phylogenetic tree, would make the figures more impactful and easier to interpret.

Response: We are grateful for the suggestion. We have added a more detailed interpretation regarding Figure 1 and Figure 2. More detailed statistical analysis was added on Table2.

3.The first time an abbreviation is used, such as "ROS" for reactive oxygen species, it should be spelled out in full with the abbreviation in parentheses. Ensure all abbreviations are defined when first introduced.

Response: Thanks for your careful checks. Based on your comments, we have made the corrections in the whole manuscript.

4.The discussion is comprehensive, but integrating more recent studies would provide a better context for the research.

Response: We agree that more research background would help to understand the details of the interaction and enhance the richness of the article. We have added line 379 and line 389.

5.Check the formatting of the references section to ensure consistency with the journal’s guidelines. Pay attention to punctuation, italics for species names, and correct citation order.

Response: Thanks for your careful checks. We have made the corrections to make the journal’s guideline harmonized within the whole manuscript.

6.Make sure all units of measurement and symbols (e.g., μg/mL, °C, p-values) are consistently formatted and follow the same style throughout the manuscript.

Response: Thanks for your careful checks. We have made the corrections to make all units of measurement and symbols harmonized within the whole manuscript.

7.Mention the software and version used for statistical analysis or any other computational work. This adds to the reproducibility of the study.

Response: We have rewritten this part according to the reviewer' s suggestion.
